# Reflecting on Teaching Practice: Adopting Islamic Liberatory Pedagogies within Muslim Institutes of Higher Education in UK (MIHEUK)

Imran Hussain Khan Suddahazai

Centre for Islamic Finance, Bolton University, Bolton BL3 5AB, UK; ihk@suddahazai.org

**Abstract:** The aim of this paper is to discuss the practice of Islamically informed liberatory pedagogical practice within a MIHEUK, through a 'self-reflective' dialogue. As the former course leader for Education Studies at the Markfield Institute of Higher Education, the paper examines the derived reflections and experience of the author teaching the BA degree program in Islamic studies with Education and the MA Post-Graduate Degree program in Islamic Education (2019–2022). In contributing to the discussion, the paper adopts a critically reflective interpretivist–hermeneutical methodology, whereby Fazlur Rahman's double hermeneutical model is utilised to contextualise core educational principles from the Islamic weltanschauung (1982). The paper cites an example exercise study utilised to analyse the adoption of Islamically informed liberatory pedagogical practices. The findings from this exercise reveal that the students' general ability to critically reflect upon the Islamic educational tradition reflect the influence of the teacher. However, the increased ability of the students to act independently and think critically to challenge the limits to their own potential or understanding of the sources dictating their religiosities and subjectivities is from a place of authenticity, thereby demonstrating the transformative nature of self-realisation as *conscientization* (critical consciousness) and its realisation of latent potentialities.

**Keywords:** Islamic education; higher education; pedagogy; conscientization; liberation theology; critical thinking; critical literacy; constructivist; double hermeneutics; interpretivist



> "Universal teaching shouldn't be placed on the program of reformist parties, nor should intellectual emancipation be inscribed on the banners of sedition. Only a man can emancipate a man". (Rancière 1991, p. 101)

## 1. Introduction

The *Insan*, in its designated status of *"khalifah fil 'ard"* (Qur'ān 2:30)[1] is an agency of Divine expression and instruction. The Islamic worldview considers *humanity* as a distinctive creation whereby individuals are fundamentally considered to be equals but are subjected to *"trials and tribulations"* (Qur'ān 29:2) to determine their strength of character and more prudently to teach them through direct experience. The response (actions) of individuals to these challenges and teachings defines their status and station. In accordance with classical Islamic scholarship, individuals striving to attain and act with virtuous knowledge are elevated by Allah (swt)[2] to stations of honour and recognition in formal or informal positions. The individual symbolised as a microscopic representation of the macrocosm (Al-Attas 1980) possesses a finite limit of the attributes of Allah (*sifat Allah*) as latent potentialities to be realised. The distinguishing feature of the teachers and students is the depth and degree to which they have realised these latent potentialities. Realisation emerges from a systematic process of learning to explore, investigate, and discover through a dialectical approach. This entails a syncretical, as opposed to a conflictual perspective, on dualistic phenomenon, thereby leading to the harmonisation of theory and practice (Al Zeera 2001). Ostensibly, contrary notions of absolute or relative reality, spiritual or material existence,

and the eternal or temporality of creation are reconciled as paradigms of a single reality. This approach, therefore, prevents the fragmentation of the self, thought, values, norms, and the concept of isolation or asceticism from the world. The notion of the other is, thus, not one of conflict, but of another dimension that must be comprehended before a holistic perspective is formulated. The Qur'ān stipulates, "*Allah could have made everyone the same, but He created diversity in society so that man could learn from the divergent opinions*" (Qur'ān 5:48). Thereby, once comprehension reaches a critical level of understanding, which is marked through 'action', it is subject to change and transformation. A central aim of this paper is to understand this constant change and transformation in the human lived experience through the practice of pedagogy. This can be conceived as the process by which theory and practice are constantly evaluated and redeveloped to address the contemporaneous context, so that learners are able to apply new methods for understanding and analysis. This realisation leads to the unifying principle of *tawhid*, a holistic perspective and understanding on a matter, situation, or circumstance.[3] This epistemological position leads to a greater appreciation for the appellations of the faith, the *din of Islam*, as a direct corollary to the evolution of a nomadic pastoralist society into a universal civilisation. This is proportionally reflected by the intellectual and spiritual development of the individual human beings comprising it. Hence, the *din of Islam* is considered a broad civilisational endeavour that is reliant upon the cultivation and nurturing of human potential to realise its creativity. This is personified by the notion of "*imam (faith) becoming intimately linked to human action (amal)*" (Sahin 2013, p. 203), whereby the declaration of *shahadat* is not simply the exclamation of a canonical testimony but one that demands the active witnessing of the faith as 'action' by the individual. From a teaching perspective, this implies that teachers functioning within the Qur'ānic ethos should strive for definitional superiority, in terms of virtue, knowledge, and realisation of latent potentialities. This will become evident from their aptitude in teaching and guiding their students towards assiduous self-development through a dialogical–dialectical process rather than the Freirean 'banking' model approach, which aptly describes the propagation of knowledge in Islamic centres of learning.

Contemporary Muslim scholarship narratives have matured to acknowledge these issues and moved to examine the functionality and purpose of Islamic educational models within Western societal contexts and the reaction against their emerging influence (Ezzani and Brooks 2019; Arar et al. 2022; Lahmar 2020a; Sahin 2013). Thematically, contemporary research developed within the last decade has focused upon such topics as policy formation; reforming existing cultures and practices; developing relationship with stakeholders; extant educational leadership models; and issues relating to gender, feminism, and social justice within Muslim faith-based schools and traditional Muslim seminaries, which includes supplementary and masajid-based schools (Maktabs) (Arar et al. 2022). Studies specifically focusing upon the practice of educational pedagogy within Muslim schools depict 'teaching' to be devoid of critically reflective, collaborative, and democratic practices (Wong et al. 2021; Lahmar 2020a; Sahin 2017). Although the research cited above provides us with new insights into the Muslim schooling experience, it does not discuss higher education. The review of literature reveals three papers that specifically address further educational issues within Muslim Institutions of Higher Education in the UK (MIHEUK). Suddahazai's two papers, one focusing on the formation of student identity through pedagogical practice within MIHEUK (Suddahazai 2021), and the other on the adoption of specific pedagogical approaches to teach Islamic, Banking, and Finance courses within MIHEUK (Suddahazai and Manjoo 2022). Whilst Lahmar's (2023) paper on "Islamic-based educational leadership in UK higher education" provides a unique reflection on the challenges and opportunities for Muslim educational leaders of MIHEUK, she discusses these in terms of policy adherence, development, and evaluation and its relationship to the accreditation, credibility, and validation of these institutions in the British context. These hybrid Muslim institutions have been established upon the framework of traditional British universities, in terms of organisational structures, administrative procedures, and academic programs, as they



have been granted the status to award accredited degrees in the UK, unlike traditional Muslim seminaries.[4]

Specifically, the aim of this paper is to focus upon the practice of educational pedagogy within MIHEUK through a '*self-reflective*' dialogue. As the former Head of Education Studies at the Markfield Institute of Higher Education (MIHE), the paper discusses the derived reflections and experience of the author teaching the BA degree program in Islamic studies with Education and the MA Post-Graduate Degree program in Islamic Education (2019–2022).

The MA program in Islamic Education was initiated by Dr Abdullah Sahin in 2010, with an exclusive focus upon Education through Islamic philosophical and methodological approaches to counter notions of indoctrination, authoritarianism, and extremism, whilst the BA program, started by his successor Dr Fella Lahmar in 2017, awarded degrees in the discipline of Islamic Studies with Education with an emphasis on its commitment to the development of analytical and critical thinking through rigorous methodological research.

## 2. Methodological Approach

In contributing to the discussion, the paper adopts a *critically reflective interpretivist–hermeneutical* methodology to discuss the use of teaching methods from the emergent field of Islamically informed critical liberator pedagogical practice to graduate students of religious seminaries within an MIHEUK. The interpretive lens provides an approach to focus attention on the social contexts to which these social actors have been exposed. As there are no predefined variables, the objective is to comprehend the students' experience, because the interpretive methodological perspective "*was conceived in reaction to the effort to develop a natural science of the social . . . Its foil was largely logical empiricist methodology and bid to apply that framework to human enquiry*" (Crotty [1998] 2020, p. 67). Instead, interpretive research perceives social reality as being entrenched within, so it cannot be disentangled from the social context, as students interpret reality though a sense-making process rather than a hypothesis testing process, whilst the adoption of critical reflection allows for the uncovering of implicit oppressive social structures and *hegemonic assumptions* (Brookfield 1995; Giroux and McLaren 1986; Kanpol 1999) through Freire's conceptualisation of 'conscientization' (critical consciousness), a 'rejection of passivity and the practice of dialogue', or 'education as the practice of freedom' (Freire 1974).

This is realised as an engagement with the 'self', to identify inherent ambiguities and conflicts in

> " . . . one's lived experience, and understanding and overcoming dominant myths, traditions, and ideologies in order to reach new levels of awareness of being an "object" in a world where only "subjects" have the means to determine the direction of their lives." (McLaren and Jandrić 2018, p. 5)

Conscientization is, therefore, a process to becoming a 'subject', collectively alongside all oppressed subjects, through the function of education as a dialogical process of struggle, praxis, and reflection.

Whilst the explicit adoption of *Reflection* in this context implies that the author, as the primary instrument of data collection and analysis, the inquirer, experiences a process of '*turning back onto a self, where the inquirer is at once observed and an active observer*' (Steier 1995, p. 163), the mind, in thinking on itself, engages itself as the subject; thereby, reflective practice simultaneously plays the role of subject that reflects an object that is reflected. "*That is, he or she becomes the object of the analysis, and it is precisely through making oneself the object of self-inquiry that a person really becomes the subject of his or her experience*" (Mortari 2015, p. 1).

Thus forth, the overarching objective of this discussion is to reflect upon the 'lived experience' (van Manen [1997] 2016, p. 35) of the students, as social actors in contexts or situations, whereby "*the details of the situation to understand the reality or perhaps a reality working behind them*" (Saunders et al. 2009, p. 111) can be uncovered. The sample of the students being discussed is varied with regards to the degree programs being studied.

For example, the majority of students I taught at the BA level were from a seminary background, as this two-year degree program was specifically aimed at Madrasa graduates, as it recognised their alim or alima[5] qualifications to be the equivalent of a first-year of a degree program.[6] The MA students, however, were from an assortment of backgrounds. This included students and teachers from the UK and subcontinental seminaries such as Deoband and Karachi, graduates of international universities in the Middle East such as Al Azhar, Medina, Mecca, and in the Asian subcontinent such as the Islamic universities of Islamabad and Malaysia, graduates of mainstream British universities such as Birmingham, Brunel, Leeds, Leicester, Nottingham, and UCL; alongside working professionals and educators, that included serving headteachers and principles of UK Muslim schools.

Thereby, the responses, engagement, and application of the students was varied in accordance with their respective degree status. The method adopted to understand and appreciate core educational terms, such as the notion of pedagogy from within the classical Islamic perspective and its eventual realisation and contemporary practice was Fazlur Rahman's hermeneutical concept of '*double movement*'. Whilst hermeneutics, as a method, allows us to move from a position of abstraction and possible ignorance to clarity (Sahin 2013, pp. 52–54), Rahman's double movement theory furthers our contextualisation of the notion of pedagogy and assists us to understand its meaning within its classical and contemporary socio-historical, cultural, and economic contexts (Rahman 1982, p. 5). Rahman's theory allowed the students to differentiate between the text, author, and the reader by focusing directly on the meaning of the text as is, rather than enforcing the perspective of individual translators or commentators. In order to attain this reading of the text, Rahman had surveyed the historicity of the Qur'ānic text, not only in terms of the *asbab nuzul* (occasions of revelation) but for the derivation of universal values, which Rahman appropriating a term from al-Shatibi, refers to as '*religiomoral*' (Swazo 2011, p. 3). Rahman differentiated these '*religiomoral*' values from explicit laws, as they can be applied in accordance to the necessitating context outside the time and place of revelation rather to specific situations or milieu as they occurred at the time of revelation or shortly thereafter, as dictated by the *asbab nuzul*. Therefore, the use of the 'double movement' was initially aimed at a non-textual interpretation of the Qur'ān, whereby one commences from the contemporary context before returning to the time of revelation of the Qur'ān (First Movement (FM)) and then proceeds back into the contemporary period (Second Movement (SM)) (Rahman 1982, pp. 5–6). The initial First Movement (FM) entails two stages: The first stage encompasses an appreciation for the implication of a Qur'ānic statement through an exploration of the historical context that the statement originally addressed. This, according to Rahman, consists of both a micro context, which can be referred to as the *asbab an-nuzul micro* (focused on particular and precise moments) and a macro context, *asbab an-nuzul macro* (condition of society, customs, institutions, and life in the Arabian Peninsula at the time of the revelation of the text) (Ainurrofiq 2019, p. 137). The Second Movement (SM) then moves to examine the sociological state of contemporary society. It undertakes this by specifically and implicitly applying the initially generated broad generalised perspective of the 'religiomoral' values derived from the historicity of the text (*asbab an-nuzul: micro and macro factors*), to the contemporary socio-historical context or situation under examination (Rahman 1982, pp. 5–6).

The students' journey through this process was recorded through their portfolio of work that included group projects, progress reports, class discussions, regular classroom quizzes, and online learning portals such as Google classroom and Moodle. This work was then periodically assessed through formative assessments such as presentations (individual and group) and class debates, alongside summative assessments, which included differing types of written and practical assignments such as literature reviews, essays, group reports, peer-to-peer assessment, and presentations.

The discussion has been organised in accordance with the adopted methodological approach, with the initial point addressed being the *critically reflective* paradigm, which introduces the concepts of criticality, liberatory theology, and their place in the Islamic

tradition. The discussion then introduces the *interpretivist–hermeneutical* method to demonstrate the realisation and experience of the students' *conscientization* as analysed through a reflective exercise. The discussion concludes with a post-reflective analysis of the exercise and adoption of Islamic liberatory pedagogies within MIHEUK.

### 3. Critical-Reflective Approach: Disclosing Experiences of Teacher Centred Pedagogy

The students from traditional Muslim seminary backgrounds enrolled onto the BA and MA Islamic education courses[7], revealed, through their portfolio of work that includes class discussions and summative assessments such as presentations, that their experiences of teaching within seminaries should be classified as teacher-centred pedagogy (Suddahazai 2021; Lahmar 2020a; Sahin 2013). As the teacher, teaching though didactic methods formulated the solitary authority responsible for educating students. A type of pedagogical approach that is also referred to as autocratic, authoritarian, or folk pedagogy (Olson and Bruner 1998). Due to the contextual role of the Madrasa institutions, the contemporary embodiments in the form of seminaries or dar-ul ulum's are inveighed with an authority amongst the adherents to guard and preserve the authentic Islamic tradition (Cherti and Bradley 2011). Thereby, those responsible for the teaching of that material are afforded venerable status with some students,[8] even though resentful at their overall educational experience with these seminaries, nevertheless, speaking of their teachers in deferential tones.

For example, a group of first-year BA students undertaking a formative group presentation upon their experience of pedagogical approaches in Muslim seminaries, entitled their presentation as: '*Teachers as heirs of the Prophets*', arguing that their seminary teachers, on account of propagating prophetic and thereby divine knowledge, were above criticism and should be respected. This further entailed descriptions that described their teachers as being akin to their parents and responsible for the development of their moral characters. In contradistinction, a similar exercise undertaken by the MA students revealed a greater maturity and balance in their perspectives, as presentations covered various nuances that were particular to the student or group. MA students that had studied aboard shared their respect for their teachers but described their roles in the learning process to have been passive, which was supported by students experience of seminaries in the UK, as students adhered to the accepted classroom behaviour and relied solely on the teacher for instruction on the content to be memorised and repeated. For instance, the students suspected that the teacher's utilisation of 'memorisation' was the only viable method, as the content to be assessed in the examinations was to be reiterated verbatim. This implied that there was no concept of personalised or group-centred teaching other than to address pre-learnt facts through *en masse* question–answer sessions at the start and end of classes, as the entire lesson time was dominated by the teachers with their teachers to be formal and detached from any emotional relationship or connection, although this feedback was not shared by students with secular educational experience and degrees from mainstream universities.

The students were describing a foundational pedagogical approach with various denominations in the literature, such as 'Direct Instruction', Didactics, and 'Active Teaching' (Good 1983). This is the most rudimentary method of teaching, as it simply entails the teacher to explain, demonstrate, and engage with the students through remember-and-recall exercises. The students conceded that their previous learning experience of teacher-centred pedagogies within traditional seminaries had made no distinctions in their abilities, learning styles, or the consequential impact upon them as learners, in terms of their understanding or application of the learnt knowledge[9]. This exemplifies the *Behaviourist* perspective on learning, which perceives a change in behaviour as merely a verification of learning as confirmed by the students through the recall of factual, explicit, and sequential learning or information through their examinations (Brown 2004; Pritchard and Woollard 2010). Therefore, the traditional teacher-centred approaches are described as passive, as the learners do not engage with the knowledge being delivered as it does not stimulate their higher cognitive processes (Blakemore and Frith 2005). These are processes through which

the learner develops ' . . . the ability to reflect on the learning experience and incorporate new knowledge with pre-existing knowledge' (Stoney and Oliver 1999, p. 26, cited in (Redmond 2014)).

Dewey (1933) perceived this as an 'active, persistent, and careful consideration of any belief or supposed form of knowledge in the light of the grounds that support it and the further conclusions to which it tends' (p. 9). Thereby, reflection becomes an 'inner dialogue, where the learner takes on both the protagonist and the antagonist roles sequentially in an attempt to reframe [their] understanding . . . [as] learners retreat from interaction to focus on individual and internal reflections' (Soo and Bonk 1998, pp. 3–6, cited in (Redmond 2014)), thereby making 'interactions meaningful, leading to changes that are referred to as learning' (Aminifar and Bahiraey 2010, p. 414, cited in (Redmond 2014)). However, in contradistinction, traditional teacher-centred approaches condition the lower cognitive processes such as detection, response, remember, and recall, to become more functional and automated (Mayer 2020; Tobias and Duffy 2009).

This form of '*rote*' learning, as described by the students and the literature, is effective and efficient in retaining information but inept in being able to transfer it with meaning (Good 2008), as learners memorise individual terms and passages of words without understanding the contextual implication of their readings.

Therefore, it can be argued that teacher-centric methods that the students experienced were devoid of genuine dialogical approaches, which could have established a relationship between the learner and the learned. Freire, for instance, argued that the teacher-centric approach hinders the 'existential development and agency of learners' as the teacher becomes central to the learning process instead of the student. Learners in such instrumental learning milieus are treated as objects of functional literacy rather than subjects, who could speak and reason through dialogue.

Freire identified this to be the '*banking concept of education*' (Freire 1970b, p. 73), whereby learners are considered as empty vessels to be filled by the depositor, the teacher, with their knowledge. Whilst Bowles and Gintis (1976), perceived it as the '*Mug and Jug*' approach, with the teacher as the proverbial 'Jug', filled with content and the learner as the passive receptacle, the 'Mug' to be filled.

## 4. Constructing Critically Reflective Pedagogies

The students' willingness to reveal and discuss their intimate educational experiences and subjectivities provides us with a further dimensional appreciation for Freire's (1974) call for 'conscientization' as a form of purging and disclosure of conditioned learning and partialities. By altering our epistemological and methodological lens, from the 'despairs and lacking' of the students to concentrating upon their lived life experiences and existing knowledge bases, helps us as educators to realise the latent innate potentialities within individual students. Deriving from the work on learner- or student-centred education as influenced by Enlightenment thinkers (e.g., Rousseau, Pestalozzi, Froebel), which situates the learner as the focal point of learning, we are able to garner the student perspective on the learning practice. This is substantiated through a methodological application, in the form of 'constructivist learning', which effectively accentuates the active role of the learner in the formation of knowledge through individual or communal learning activities.

Therefore, the adoption and use of constructivist pedagogical approaches such as cognitive constructivism, allowed us to perceive '*learning as a constructive process*', as originally conceived by Piaget, for his " . . . *fundamental insight was that learners [individuals] construct their own understanding*" (Woolfolk 2006, p. 38), so as Piaget identifies, learners must become agents of their own learning processes. The practical expression of constructivist thought can be gleamed through social-constructivist theory, which entails " . . . *negotiating understanding through dialogue or discourses shared by two or more members of the community of people who are pursuing shared goals*" (Brophy 2002, p. 1). This emphasises the function of language, culture, and social interactions on cognitive development, inferring that language serves as an instrument for shared interactions through dialogue between learners as active agents

and their peers and teachers within a cultural context. Subsequently, according to Jarvis (1987), this collaborative approach allows for the development of higher cognitive functions such as thinking, reasoning, and problem solving. For instance, through classroom discussions and preparation for group-orientated assessments, the students within the BA program were encouraged to utilise their own educational experiences and knowledge of Islamic educational history to explore the broad perspectives on pedagogy.

To do this, students were introduced to literature related to discussing the literal interpretation of the term pedagogy[10] as '*to lead a child*', from the classical Greek *paidagogas*, '*pais*' (*paid*), the child, and '*agogus*', leader. Thus, it became apparent to all of the students that this is, at once, a derived practice and a societal role from the Greco-Roman civilisation, whereby a *paidagogas*, a servant [slave], was entrusted with the education of the children of wealthy and influential families and has subsequently in the contemporary era become associated with the specific notion of teaching and teaching methods, especially within educational institutions (Waring and Evans 2014; Leach and Moon 2008). After extensive reading, all the students in the BA program were also able to appreciate that there existed no overarchingly agreed-upon definition for the term pedagogy. Thereby, the literature on the topic is focused on the ways teachers can adapt their teaching methods and learning strategies to the intellectual capacity and individual needs of their students (Alexander 2008). Furthermore, this broad conceptualisation of pedagogy as a field of study was organised to reflect the prevailing knowledge paradigms of the social sciences. These approaches to pedagogy incorporate the positivist or traditional functionalist methods, such as Teacher-Centred Learning (Simon 1985, p. 77); the interpretivist perspective based upon ideas of Learner-Centred approaches (Alexander 2008); and the critical dimensions, focusing upon empowerment and transformation (Waring and Evans 2014). Having gained an appreciation for the Greco heritage of pedagogy, all of the students[11] were readily able to identity with the Islamic perspective on the foundations of 'pedagogical practice' and perceived it as being encapsulated within the methodology of revelation utilised by Allah (sbt) to '*guide*', '*nurture*', and '*teach*' Muhammad(s) just like the prophets that preceded him (Qur'ān 3:159; 41:6; 68:4). They were able to recognise that the first five revealed verses (Qur'an 96:1–5) contain the "*concepts of reading, learning/knowing and the pen . . . [demonstrating] significance of uttering and writing the revealed scripture . . . [which] . . . is locked in the very nouns that designate the Qur'ān*" (Haleem and Haleem 2001, pp. 1–2). This demonstrates the hidden qualities and insights possessed by traditional seminary students, as most of the students were polyglots and could actively engage 'with' and 'in' the class. Dare I say it, the classroom became a 'platform for the expression of their learning', as students were eager to demonstrate their linguistic prowess and efficiency through presentations and discussions on Qur'ānic Arabic and its intimate relationship to education. For example, Sahin's (2013) '*cloud-grass*' theory of Islamic education was introduced to the students as an example of contemporary Muslim scholarship and development in the field. Sahin advocates the use of the term *tarbiyah*, a method that entails the "*gradual, stage by stage developmental process informing an organism's growth until the complete actualisation of its potential*" (Sahin 2013, p. 182). In the application of Rahman's (1982) hermeneutical *double movement* approach to the concept of 'tarbiya', it becomes apparent that the concept of 'critical pedagogy' and *reflective practice* share similarities to this classical Islamic understanding of education as 'tarbiya' (Sahin 2013). This implies that 'leaders' as teachers or educators are demonstratively familiar with the extensive nature of pedagogical methods and their appropriate applications in accordance with the contextual circumstances and situation. Thereby, educators with intimate theoretical and practical knowledge of the subject matter can initiate a perpetual dialogue, which begins with themselves, as a form of self-reflection and with their students, to gauge their learning experiences.

## 5. Discernment and Contemplation

A close examination of these 'Islamic' ideals reveals that they find voice within the critical pedagogical tradition, in that they are conceived as the methodological gateway

towards the anticipated societal evolution, initiated through the agency of the enlightened individual, in the form of an intellectual critical teacher with the ability to transform students and followers through logical rational discourse (Gore 1993, p. 121). Additionally, critical pedagogy is "*a way of thinking about, negotiating and, transforming the relationship among classroom teaching, the production of knowledge, the institutional structures of the school and the social material relations of the wider community, society and nation-state*" (McLaren 1999, p. 454), implying that critical pedagogy engages with the practices of justice, democracy, and ethics within educational contexts, as it considers the modern education system to be a source of power and influence in maintaining the authority of the dominant capitalist *bourgeois* class. Educational institutions imbue students with "*a system of values, attitudes, behaviours, beliefs and morality that supports or reproduces the established social order and the class interests that dominate it*" (Braa and Callero 2006, p. 358). This is usually attained through such measures as the hidden curriculum to condition students to conform to the hierarchical structures of power. Thus, students eventually mature into submissive, diffident members of society, as part of a tractable labour force with limited socio-political engagement and influence.

However, there is a sharp distinction between the faith-centred epistemological foundation of the Islamic and the secular humanistic vision of critical pedagogy such as the social vision on critical education or the 'Instructional' (Shor 1987, pp. 215–31) which perceive educational institutions, within a limited capacity, as simple agencies for social and economic mobility (Duncan-Andrade and Morrell 2008). Developed largely under the aegis of neo-Marxism, the critical theory of the Frankfurt School, and postmodern social theory, these ideals profess an epistemological underpinning that rejects the idea of axiomatic universal knowledge and truth and instead seek to "*to regard [those with religions convictions] as quaintly pre-modern or as the needy recipients of our saving (an ironic word) wisdom*" (Fish 2005, p. 1).

Thereby, this presents a contention with regards to a crude adoption of critical pedagogy as a teaching method to Muslim faith-orientated students within MIHEUK, as it disregards faith-centred methodologies as viable lenses for the analysis of power, authority, and influence based narratives.

The resolution to this epistemological dilemma is providentially unearthed from a brethren faith tradition. Gustavo Gutierrez's ideas on social justice through his ' . . . *theology of liberation*' (Brown 1990, p. 13) implores Christians from all social classes to act upon the teachings of the gospels to eradicate oppressive class structures. Gutierrez, a Peruvian Catholic priest, articulated a theological vision that espoused a prophetic Biblical tradition as symbolised in the personhood of Jesus and encapsulated in the institution of the church to be based upon the struggle for human values and dignity. Omar (2008, pp. 93–94) discusses it as a form of liberation against colonial and post-colonial policies, attitudes, and structures of power, authority, and influence. The concept of liberation is taken to be the anthesis of oppression and, therefore, becomes a demand for 'voice', which translates as self-determination in socioeconomic, political, intellectual, and spiritual terms (Ibid). This, Gutierrez identifies, is due to God revealing Himself " . . . *only in the concrete historical context of liberation of the poor and oppressed*" (Brown 1990, p. 67), the implication being that God is found 'in amongst' the struggle of the insignificant, the destitute, the poor. This led Gutierrez to argue that to find God, the Gospels provide epistemic privileges to the oppressed, as it addresses those " . . . *who have not yet named the world, the marginal, the silenced, the defeated* . . . " (Welch 1985, p. 34). Thereby, literacy becomes an instrument of liberation, as teaching the oppressed to not only read, but to read the scripture in a method that proclaims their humanity[12], fulfils Jesus's prophetic responsibility, in that "*He has sent me to bring glad tidings to the poor, to proclaim liberty to captives* . . . " (Bible Luke 4:18–19).

This implies that 'liberation theology' is a theological conviction that it is the " . . . *destiny of humankind to join with the creator in* . . . *building a world that is both just and free*" (Jarvis 1987, p. 31), due to the independent will of the human being in choosing to work with God.

Freire (1974) suggests that for this to transpire, the process of conscientization is required, whereby the oppressed become aware of their situation and circumstances and seek to transform themselves. Thus, Gutierrez adopts Freire's conscientization as "*a pedagogy to animate a new theology*" (Lange 1998, p. 83), by ensuring that there is no distinction between theory and practice, [praxis: action and reflection] as truth is something to be done.

Freire (1970b), therefore, conceives of critical pedagogy as praxis in the form of '*problem-posing education*', which responds to the '*essence of consciousness*' of individuals as '*conscious beings*'. This forces individuals to think critically about their role in the world. They learn to perceive the world as a transformational rather than a stagnant predictable phenomenon. Teachers work with students to disseminate knowledge and engage them incrementally to eventually lead discussions and establish a self-realised form of further learning (Duncan-Andrade and Morrell 2008).

In essence, this justifies the adoption of liberatory critical pedagogical approaches to the Islamic praxis of teaching as contemplating upon research from the prophetic traditions, which leads to a "*sense of the larger context, the larger forces that shape and mould not only who we are but our projection of where we want to go*" (West 1993, p. 227).

## 6. Reflection and Liberation

It could be rationally postulated that the conceptual notion of liberation theology is encapsulated in the heritage of the Islamic tradition. Just as Jesus had become the 'word of God' (Bible Revelation 19:13), Muhammad inspired by the commandment to *read*, was transformed from the mundane, ordinary to the supreme, the divine, by the virtues of struggle and strife. Consequently, the discernible Islamic concept of teaching and pedagogy, as derived critically and empirically from its sacred text and tradition, is founded upon the conduct of personalities chosen to become vessels and embodiments of the divine message. These Prophets (*nabi*) and Messengers (*rasul*) were developed as imams to lead their societies 'out of *Jahalia*' (Qur'ān 7:199) towards truth (2:109; 2:213; 2:252; 4:170), certainty (15:99), and faith (16:123; 4:125) by teaching them to realise the purpose of their existence (51:56).

Prophet Muhammad(s), acknowledged as the seal of the Prophets *(Khatam al nabiyin)* (Qur'ān 33:40), is considered by the Islamic tradition to be the ultimate personification of the teacher leader ideal. In Qur'ānic terms, this implies that in the personhood of the Prophet "*is an excellent example (uswah hasanah) . . .*" (33:21), or '*role model*'. The designation of Role implies " *. . . a person's characteristic or expected function . . .*" whilst a "'*Model*' is a person who is regarded by others as an outstandingly good example of a particular role*" (Adair 2010, p. 1).

In applying this understanding of role model to "*God, the Prophets, the saints, and religious teachers . . . it might be termed a 'Sacred Other . . . from whose perspective a believer sees themselves and perceives the discourse for creating and performing oneself*" (Gregg 2005, p. 112). In examining this conception through the lens of Max Weber's (1962) Charismatic model, we can discern the acceptance of the authority of the Prophet(s) due to his personal '*charisma*'. Weber perceives this to be " *. . . a certain quality of an individual personality by virtue of which he is set apart from ordinary men and treated as endowed with supernatural, superhuman, or at least specifically exceptional qualities*" (Weber 1962, p. xviii).

This implies that the charismatic personality of the teacher resonates with its students' need for divine intervention in countering their social contexts. Therefore, the emergence of the *ummah* (Muslim community) can be idealised as a direct reflection of the personal charisma of Muhammad(s), "*imitatio Muhammadi*' (Schimmel 1975, pp. 144–58), as " *. . . charisma becomes a creative and revolutionary social force in society*" (Takim 2006, p. 3) to challenge traditionally accepted structures of power, authority, and influence.

However, this does not place the Prophet outside the experience of the common people's 'ordinary' and 'mundane' secular life. On the contrary, the Prophet's lived experiences (*sunnah*) demonstrate his engagement with the world. The examination of the sunnah, as

an active living tradition ([Rahman 1982](#)), alongside the methodological development of the hadith literature, reveal that they not only capture the essence of the prophet's charismatic personality, but are instigators for the development of a dynamic tradition of charismatic Muslim scholars and teachers. Consequently, the underpinning for pedagogy is derived from the notion of the '*uswah hasanah*', as it embodies the "*art of being human*" ([Sahin 2013](#), pp. 168–69). The Qur'ān defines Muhammad(s)'s primary role as a teacher (Qur'ān 62:2) with exceptional character (68:4), sublime conduct (33:21), a tempered approach that endeared his followers to listen (3:159) whilst helping them to reflect (34:46) through guidance towards a true understanding so the followers can also bear witness (12:108; [Suddahazai 2015](#)). Broadly applied, this honour and status associated with the role-based leadership model of the Prophet 'teacher' in Islamic thought is extended to society, as exemplified by the Qur'ānic injunction to leave behind a contingent of individuals during times of war, so that they would become vessels to continue, preserve, and propagate the teachings (Qur'ān 19:122). Furthermore, this approach to education is empirically demonstrated by the Prophet, after capturing seventy prisoners during the battle of Badr (13/03/624). As these men were all literate, the Prophet(s) declared that if one prisoner taught ten Muslims the art of reading and writing this would act as their ransom and, thus, they would be free. Thereby, the difference between the functionality and roles of prophets and teachers is in degrees, as teachers, in the Islamic tradition, are described as the *heirs of the Prophets* and " . . . *anyone who desires to attend the assembly of the Prophets should visit the assembly of the teachers*" ([Rahman 2005](#), pp. 91–92).

This highlights the transformative aspect of the Islamic perspective on liberatory pedagogical practice as it " . . . refers to individuals who recognise the existing needs of potential followers, but go further, seeking to satisfy higher needs to engage the full person of the followers in terms of Maslow's hierarchy of needs" ([Burns 1978](#), p. 4). Therefore, the teachers, as the heirs of the prophets, seek to support the transformation of their students by meeting their higher needs such as self-esteem, self-fulfilment, and self-actualisation. Transformation is only attainable, however, as a consequence of the teachers' and students' ethical and moral interaction, in that the resultant change is dependent upon the satisfaction of the authentic needs of the students. Therefore, the concern is not orientated towards the scrupulous analysis of pedagogical methods but rather " . . . is concerned with what knowledge(s), ways of thinking, dispositions and subjectivities are actually (re)produced in/through particular pedagogical encounters" ([Tinning and Bailey 2009](#), pp. 18–19).

## 7. Sense Making through Critical Liberatory Pedagogies

In discussing the experience of the '*actual pedagogical encounters*', it has to be appreciated that due to the shared Muslim identity, heritage, and fundamental religious education of the students and teacher, there existed a pre-pedagogical relationship founded upon faith-based values.[13] These inherent factors[14] led to students from traditional and secular educational backgrounds on both the BA and MA programs to demonstrate overt respect and obeisance to myself as the teacher. Therefore, this general attitude of the students facilitated the teacher's planning, design, and implementation of the program and the pedagogical approaches utilised to undertake specific objectives. Although this was a significant realisation, it also presented a slight conundrum, the reason being that the underlying purpose of both the Islamic worldview and liberation theology is to liberate or transform the individual. This implies that the individual is able to think critically and act independently through their own volition to arrive at a judgement.

Therefore, the dilemma I was attempting to resolve at the end of my first semester teaching a new cohort of MA and BA students was centred upon notions of authenticity. A question I posed myself asked if the improved developments in the students' ability to critically reflect upon the Islamic educational tradition was merely a reflection of my own teaching contribution and guidance or was it due to a Kierkegaardian notion of authenticity, as a genuine change and transformation in behaviour due to the students' own choice or exercise of free will due to their experience of the real world?

### 8. Reflecting on Practice: An Empirical Example

To address this question, prior to the start of the semester, I had duly noted the religiosity and subjectivity of all the students through Sahin's Muslim Subjectivity Interview Schedule (MSIS) (Sahin 2013, p. 69; Suddahazai 2015).

This measuring of religiosity implied the deciphering of the student's personal attitudes towards their 'belief in' and 'practice of' Islam, whilst 'subjectivity' was the level to which this has been informed by critical or reflective practices (Suddahazai 2015, p. 121). Thereby, I had pre-planned this exercise to garner initial data on the students and simultaneously establish a reference point for each student with regards to their practice of subjectivity. Furthermore, this informal anecdotal approach would allow me to gain some appreciation of the impact the course was having on the students and allow me to react to each individual requirement. Although this exercise was methodologically sound, it was not intended to be a published scientific study; rather, it was designed as an instrument for self-evaluation. Nevertheless, the exercise yielded informative observations and reflections on the transformative process of students.

The initial results of the MSIS undertaken at the start of the semester identified every individual in the BA class to be demonstrating 'foreclosed mode of religiosity and subjectivity', which implied that although they were committed to the devotional ritualistic and pietistic aspects of the faith, they had not undertaken any form of exploration to understand the rationale of the core principles of the faith. The MSIS readings from the MA class were much more varied, with students having attended seminaries in the UK and abroad, alongside those who attended Islamic universities in Medina, Mecca, and Al Azhar demonstrating 'foreclosed modes of religiosity and subjectivity', whilst students with broader experience of professional and secular education demonstrated 'achieved' modes of religiosity and subjectivity', which indicated that their religious commitment was founded upon an informed process of exploration and they were willing to question core principles to understand their functionality and purpose in the contemporary world.

These data allowed me to monitor the engagement and performance of those students deemed to be 'foreclosed' and work closely with those deemed to be 'achieved' by introducing them to more complex discussions and narratives from both the Islamic and secular traditions. Thereby, I developed a crude method to gauge the extant development of individual students, alongside recognising and appreciating their latent potentialities and limits. Therefore, the improvements in the students' performance being referred to above is with regards to their move towards more exploratory, achieved modes of religiosity and subjectivity.

### 9. Exercise Synopsis

My investigation sought to uncover the factors for the students' increased engagement and informed perspectives. I questioned whether it was due to my direct influence and instruction or the '*conscientization*' of the students awaking to their own developmental needs.

In order to address this question, I decided that I would involve all of the students in both the MA and BA classes to help me prepare a presentation for an upcoming international conference at the end of the year.[15] Working within the ethos of Islamically informed liberatory pedagogical practice, I invited all of the students to four intra-departmental sessions on research planning for the conference.

#### 9.1. First Session

The first session served to introduce the aims and objectives of the conference, which included informing the students about our collective roles in the remainder of the meeting and the subsequent three sessions. After informing the students of the task, I was to become detached from active participation and would become an independent observer of the proceedings and share my feedback in the final fourth session. The collective body of the students was delegated with the authority to decide upon a presentation topic and develop an argument in the form of a dialogical narrative to support its selection.

After I had vacated the platform at the front of the lecture hall and made my way to the side, there was for a few minutes an initial vacuum of authority and direction. However, eventually, a senior MA student, Adam[16], humbly assumed a position by standing up to address the entire cohort of students. He requested that the group decide upon a strategic course of action by firstly understanding the objective of their task and then organise themselves to address its requirements. It appears that due to Adam's credible relationship and reputation with the student body, his words stimulated reactions from the rest of the MA cohort, who agreed with his suggestion and some MA students suggested that Adam should act as the lead for this project. Although Adam did not make a case for such a course of action, the BA students also supported this assertion when asked by some of the MA students, with one female BA student revealing that they, as a group, 'did not feel they had the knowledge or the understanding to be able to lead on a project of such magnitude and as the MA students were their superiors in knowledge it would only be appropriate for Adam to lead'. Consequently, a group decision was made to appoint Adam as the 'director of the research', which appeared to correlate to my earlier identification of his 'achieved' mode of religiosity and subjectivity. Adam was currently working as the principal head of an educational organisation and held a previous post-graduate qualification from a major British university, alongside theological training from a Muslim seminary abroad. Therefore, the students had also recognised his achievements and abilities and worked with Adam to establish themselves into groups, which would be led by three MA students, primarily due to their senior educational status and the reluctance of the BA students to lead. This resulted in the establishment of around three groups of five individuals and ensured that BA and MA students would be working together as a group to firstly understand the aims and objectives of the conference and then to engage creatively to develop their ideas for a topical presentation. Adam then suggested, in agreement with the group, that the subsequent sessions be held every two weeks of that semester.

*9.2. Second Session*

The second session led to the various presentations by the three groups, but they all focused on a similar topic, the role of liberatory pedagogies in the Islamic tradition. They argued that the one of the aims of the conference was to explore the points of convergences between the traditional Madrasa and the modern hybrid academy such as Markfield Institute of Higher Education (MIHE). Therefore, they argued that the modern hybrid academy should adopt the classical Islamic approaches to liberatory pedagogy as a point of convergence between the two types of institutions.

This dialogue revealed that the group leaders, all part of the MA course, had spent time discussing the task and objectives of the conference. The BA students shared in the session that they had agreed to this decision because they understood the value of the argument presented to them by their group leaders. Although the students from the BA cohort were all identified as being 'foreclosed', some of them were now demonstrating signs of exploration and achievement by actively questioning their own understanding of the Islamic tradition. This implied that they did not defer their decisions to the leaders of the group due to diffidence, disengagement, or conformity with the process as described by the Asch Conformity Experiments,[17] but a realisation of the limits of their own understanding.

It became apparent that those students that had been identified as having 'achieved' modes of subjectivity and religiosity within the MA class, around five students, were also actively engaged and contributed the most to the discussion. The rest of the MA students that were initially identified to be 'foreclosed', also supported the arguments of their particular group leaders, and further encouraged the BA students from their groups to open up and share their points of view. This highlighted an important point for observational analysis, in that the initially identified 'foreclosed' MA students demonstrated aspects of Islamically informed liberatory practice that I had not considered, nor sought to measure in any capacity beforehand. They overtly demonstrated what Nel Nodding's describeNoddings described as the 'ethical caring nature' of pedagogy as a human practice.

Thereby, the actions of these specific MA students were consummate with their religious knowledge and lived experience of the teacher–student or peer–peer relationships.

Adam, speaking on behalf of the entire cohort, synthesised their understanding of the aims and objectives of the conference and the topic they sought to discuss. This was to be centred upon a discussion on Islamically informed liberatory pedagogical practice as the point of convergence between the traditional classical Madrasa institutions and the new hybrid academies in the UK, such as MIHE.

Adam then allowed another MA student, Nur, a high-performing international female student identified initially as demonstrating the 'achieved' mode of religiosity and subjectivity, to address the entire cohort. It was also clear that this was a pre-planned event, as Nur had prepared a presentation. She began by further reiterating her support for the idea and argued for the adoption of a framework for the development of an argument to support this narrative. She suggested that we examine the work of Farid Esack, the activist, educator, and pioneer of the liberatory field within Islamic and Muslim discourse. She had been introduced to his writings through an elective Islamic Studies module under the supervision of Dr Haroon Bashir.[18] This demonstrated her ability to work in an interdisciplinary fashion and was appreciated by all of the MA students, as they realised the cross-fertilisation of ideas in practice. Nur was able to engage with complex ideas and explain them to BA level students with foreclosed subjectivities and religiosities, thus demonstrating her own ability for critical thinking, reflection, and teaching. She introduced Esack's ideas on Qur'ānic hermeneutic of liberation and pluralism to conceptualise the Islamic liberatory narrative within the context of state-sanctioned apartheid, racism, and discrimination in South Africa, which Esack compared to the socio-economic and political cultural landscape of 7th century Mecca in which the Qur'ān was revealed. Furthermore, Nur was able to identify and cite his definition of Islamic liberation as "*one that works towards freeing religion from social, political and religious structures and ideas based on uncritical obedience and the freedom . . .* " (Esack 1997, p. 83) as being similar to Gutierrez's conception of the ' . . . *theology of liberation*'. She concluded that Esack also believed that this could only be attained through the notion and nature of 'jihad', which should be conceived in the contemporary context through the lens of Islamic liberation as " . . . *struggle and praxis . . . an effort, an exertion to the utmost, a striving for truth and justice.*" (Ibid.)

At the conclusion of Nur's presentation, the rest of the students did not query her with regards to her understanding of Esack's narrative or challenge her with alternative models. This appeared to demonstrate the students lacked broad awareness and knowledge of the subject field as opposed to not understanding the concept of Islamically informed liberatory pedagogies. This was made evident by a significant contribution from a male BA student, Khalid, who suggested that we should re-examine the role of the four Imams of the Sunni madhab from the lens of liberation theology. This was a significant moment as it demonstrated active and critical engagement by a student renowned for his reservation and timidness. Additionally, it provided some credence to the scaffolding techniques I had been utilising to pair students from MA and BA traditional seminary backgrounds with MA students with achieved modes of religiosity and subjectivity.

This suggestion by Khalid was received with aplomb by the rest of the cohort and it was collectively agreed under the aegis of Adam's directorship that the students in their existing four groups would research and present their ideas on each of the imams in the penultimate third session.

*9.3. Third Session*

The third session began in an orderly fashion and was self-managed and organised, thus demonstrating the respectful and responsible attitudes of all the students. All the groups were tasked with reflecting upon the early Islamic tradition through the lens of an Islamic-informed critical pedagogical approach. Each group then proceeded to present their findings through the deployment of various pedagogical methods to demonstrate their understanding of the methods. For instance, the first group utilised a lecture approach

(teacher-led), whereby a single member of the group presented the findings. This group chose a female BA student, Aaliyah, to present these findings, as she had demonstrated remarkable confidence and articulation in her contributions to the group discussions in the preceding sessions.

Aaliyah argued that Malik Ibn Anas' refusal to allow the first Abbasid Caliph, Al-Mansur, from sanctioning his text Al-*Muwata* as the official book of law of the Abbasid state, and its adoption throughout the Muslim provinces, was a sign of his awareness of a supreme form of justice and equality, as mandating the text would amount to tyranny and injustice by ascribing upon him sole authority for interpreting Divine law, to which his response was that '*people should not be forced to follow any particular school of Islamic legal opinion.*'[19] She then reflected on several contemporary scholars and their moral and ethical conduct with regards to compromising fundamental Qur'ānic values for wealth and fame.

The second group chose to share their ideas through a group presentation. The entire group presented aspects of their research and argument, with each member contributing to the debate and then being questioned afterwards by the rest of the cohort. Three students led by their MA student group leader presented on the case of Abu Hanifa. They analysed several prominent examples from his life and identified his defiant streak, which they felt was directly applicable to the contemporary times, through his response to those who disputed the authenticity of his work on the grounds that "*the followers did not do such things*". He retorted:

> "What comes to us from the Allah the Almighty is held the most supreme by us, and what reaches us from the Messenger of Allah (pbuh) we (simply) listen and obey, and what reaches from the Companions . . . we chose the best from their opinions . . . and what comes to us from the opinions of the Followers (tabi'un), so they are men like us". (Cheema 2017)

They noted that due to his defiant and critically reflective approach, he was subsequently imprisoned and tortured on refusing the chief Qadi or judge position on several occasions. In comparison with the contemporary times, the students found it difficult to fathom a scholar refusing such authority, power, and influence.

The remaining two members of this group, comprising of MA and BA students, identified the struggles of Imam Shāfi'ī to be akin to his predecessor Imam Abu Hanifa'. They claimed that he was also the victim of political persecution and manoeuvring by citing evidence of his arrest and charge for political interference in Yemen, which was eventually dismissed by Haroon-al-Rasheed, the caliph of that time.

The third group utilised a creative form of expression by adopting role-playing to enact the infamous imprisonment and torture of Ibn Hanbal during the mihnah (*inquisition*) of the 'Abbàsid caliph al-Ma'mün in 219/833. The group's narrator, a male student from the BA class, demonstrated the groups' broad awareness and appreciation for methodological analysis by citing Fazlur Rahman's 'double-movement' hermeneutics to declare that Ibn Hanbal's treatment signifies the essence of the 'pedagogy of the oppressed' thesis in Islamic discourse. The group depicted, through some comedic role-play, ibn Hanbal's (d. 241/855) refusal to accept the createdness of the Qur'ān, which resulted in him languishing in prison for over two years. The students were keen to portray this as an act of revolution, as it eventually resulted in the end of the *mihnah* and the reneging of the creed for the creation of the Qur'ān.

This presentation involved every individual in the group, and it can only be assumed from anecdotal evidence that certain members of the group living in close proximity met over a weekend to prepare the choreograph for the presentation.

The students led by Adam briefly discussed the revolutionary nature of the early classical Islamic scholars before agreeing to work within their groups to finalise their argument for the concluding fourth session.

*9.4. Fourth Session*

The fourth and final session witnessed the discussion on the conclusive final argument framework to be delivered at the conference. This final summation of their research was undertaken by the MA student leaders of the three groups. Nur began by arguing that despite enduring horrific persecution and ordeal, the struggle of these early scholars translates into a blueprint for the birth of a great tradition of liberatory scholarship and learning within the Islamic tradition. The very notion of adversity and hostility in societal contexts creates the need and the opportunity for the emergence of a dynamic culture of cross-collaboration and -fertilisation, informed by its purposeful inevitability for liberation, a response to the tyranny, injustice, oppression, and dehumanisation of the *mustad'afun* (people considered to be of inferior social status) (Esack 1997, pp. 100–1). In support, Hasan, an MA student with previous educational qualifications from a British university, was able to add to the discussion and dialogue by pulling in material he had engaged with in other classes. Hasan shared that the Islamic liberatory message had maintained its consistency and application throughout Muslim history, from addressing the tyranny of the early Meccans, to the struggles for liberation from colonial oppression during the nineteenth and early twentieth centuries. This narrative was continued by the final student leader, Sira, a graduate of a British university currently working as a teacher within a mainstream school. She was able to cite the work of scholars and reformers such as Asghar Ali Engineer and Maulana Mawdūdī, pioneers of the modern adoption and use of the term 'liberation', in its deployment as 'Qur'ānic liberation' (Engineer 2001, pp. 21–30), in their responses to British imperialism and colonisation of the Indian subcontinent some thirty-five years before Gutierrez's reflections on and in South America.

Furthermore, Sira introduced the work of more recent examples, such as the great Iranian reformer and ideologue of the Iranian revolution, Ali Shariati (d. 1977) and Farhang (1979). She identified his message to embody, in Freirean terms, the struggle against an oppressor that is also being oppressed, in this case, the American-bolstered despotic autocratic Shah of Iran. She recognised that Shariati's response was to stimulate the public by, first and foremost, transforming their thinking to perceive this as a battle between *haq* (truth), as characterised by the believing faithful public and those being oppressed and deprived, against *taghut* (falsehood), as personified by the Shah, his regime, and supporters as the oppressors and usurpers of rights and wealth (Shariati 1979, pp. 11–33).

The three group leaders had presented their viewpoints on the heritage of liberatory pedagogical practice in the Muslim tradition as a coherent single narrative. This demonstrated the in-depth engagement of the students and their internal motivation to contribute a significant argument to the conference. This was supplemented by Adam's concluding contribution to the session, which discussed the supporting pedagogical approaches to be adopted by modern hybrid academies. These Adam identified from the Muslim and contemporary secular traditions to encompass the Socratic type of questioning for clarification of understanding, classroom discussions on the assigned, and student-selected readings and texts alongside the establishment of a space and time for the students to collaborate and explore the text and narratives. Adam argued further that assessments should not measure literal understanding of the readings or factual memorisation of history but appreciate the ability of students to derive, interpret, and construct a new understanding and narratives around the lives of the historical figures they were examining.

I resumed my participation at the end of this fourth session and shared with the cohort my feedback on their conduct and performance. I revealed to them that the objectives of this exercise had been twofold, in that firstly, it sought to introduce and familiarise the students with the modern educational research process. Thereby, students gained theoretical and practical knowledge of the administrative, organisational, and procedural roles involved in developing new research. Secondly, the academic aim of the exercise was to provide the learners with the tools to conduct a critical intellectual rendition of their own heritage. This formulated a foundational experience for these aspiring teachers to explore, through

hermeneutical discovery, the art of pedagogical practice as embodied by its most sacrosanct texts and personalities with considerations to the contemporary context.

*9.5. Post Reflection*

A précised contemplation on the exercise reveals this to be an 'exploration and analysis of the narrative between the learned and the learner'. The dialogue generated from this exercise revealed the emergence of 'conscientization' within the students but in differing guises. Students identified to have 'achieved' subjectivities and religiosities, with real-world experience and exposure to a broad range of pedagogical approaches, appeared to have been awoken to their own latent potentialities. The exercise provided them with the platform to express their abilities and innate motivations to teach. They demonstrated the development of their independent intellectual prowess by arguing, in the fourth session, that the struggle for garnering 'truth in theory and gaining justice in action' was reliant upon the appreciation that knowledge was a constructed phenomenon within a specific interrelated context. They identified this to be symbolised by the existence of *dialogue*, which is directly correlated to the epistemological development of the individual and their subsequent actions (Freire 1970a; Shor 1987). This level of understanding amongst these students implied that they were bestowed upon a responsibility and trust for which they become accountable. In this case, they became scaffolds and acted as role models through a close working relationship to students identified as being 'foreclosed', thus reflecting the classical Islamic models of pedagogy. Thereby, these 'achieved' students were instrumental in transforming the views of 'foreclosed' students. The perception and image the students from a traditional seminary background had of the historical figures presented to them (four founding Imams of the Sunni madhab) was only from a pietistic juristic theological disposition at the start of the course module. However, during the exercise, the students were able to present a fresh narrative of the four Imams as social revolutionaries. In working with these students from traditional seminary backgrounds, it had become apparent that they were unacquainted with the type of content, language, culture, and methodological approach to teaching within an academic secular–humanistic framework. Through a critically reflective process entailing mutual dialogue, group discussions, and independent research, the students were able to 'self-reflect' upon their situation, demonstrating their emerging awareness or *conscientization.* The students were eventually prepared to discuss their educational experiences and adopted Freire's notion of *critical literacy,* as "*an attitude towards texts and discourses that questions the social, political and economic conditions under which those texts were constructed*" (Beck 2005, p. 2) to develop their own dialogic voice. The critical analysis of textual dialogues functioned to increase their critical self-awareness, which subsequently allowed them to reflect and think critically. The majority of the students demonstrated this increasing awareness throughout the course, as indicated by the exercise. In this case, most of the students were able to critique the influence and nurturing of their 'personalities' by the heritage and custom of the traditions they represented through their seminary education. They were able to discern that their 'reactionary defensive style of hermeneutical scholarship' (Sahin 2013) was a consequence of the method by which the *dars-i nizami* curriculum is taught by *dar al-'ulums* in the UK. They recognised through critical literacy and analysis that the *dars-i nizami* curriculum had not been forged from the scholarly and intellectual endeavours of the Muslim community in contemporary Britain, but nurtured and fostered as a response to the tyranny of colonialist usurpation of the Indian subcontinent in the 1900s. The development of these 'critical' skills also implied that the students at both the BA and MA levels were now willing to question and contemplate upon the actions of their own traditions. They further demonstrated their intellectual developments by using rhetorical approaches such as questioning the effect of the Deobandi-led reformation in India and its alteration of the *dars-i nizami* curriculum, which had been used previously to nurture diplomats and scholars through a dialectical reconciliation of the divine and secular. The MA students, in particular, were able to respond to their own query by observing that the removal of the secular perspective has

implied that Islamic education has become synonymous with training in theology and jurisprudence as opposed to understanding it philosophically and methodologically.

Therefore, in addressing the question I had posed earlier to myself, I would argue that the general ability of the students to critically reflect upon the Islamic educational tradition certainly reflects the influence of my teachings. This is evident from their utilisation of the sources, perspectives, and approaches that I had provided to them through the course. However, the increased ability of the students to act independently and think critically to challenge either their own potential or understanding of the sources dictating their religiosities and subjectivities is from a place of authenticity and demonstrates the transformation of the students through a self-realisation, '*conscientization*' of their individual need to realise their latent potentialities.

## 10. Conclusions

The discussion was constructed within the spirit of a '*self-reflective*' dialogue that sought to share the experience of a contemporary educator's practice of teaching and pedagogy within a Muslim Institution of Higher Education in the UK (MIHEUK). The reflective discussion was organised in accordance with its methodological structure, whereby the *critically reflective* paradigm allowed us to introduce the concepts of criticality, liberatory theology and their place in the Islamic tradition, whilst the *interpretivist–hermeneutical* methodology was utilised to demonstrate the realisation and experience of the students. It can be argued that the emergence of MIHEUK is due to the *conscientization* or contemplative consequence of the internal reflective experience of the pluralistic Muslim community in the UK, as the purpose and role of the MIHEUK is imperative in developing the next generation of Muslim educators and students.

**Funding:** This research received no external funding.

**Conflicts of Interest:** The author declares no conflict of interest.

## Notes

[1]   The Quranic ayahs are paraphrased utilising Quranic translations from Arberry (1955), Pickthall (1999), Maududi (1979) and Haleem and Haleem (2001).

[2]   The Arabic epithet Subhanahu wa ta'ala (swt) is added honorifically after the mention of the name Allah and is translated as "Glory to Him, the Exalted" or "Glorious and Exalted Is He". The use of swt can be derived from the following Quranic verses: 6:100, 16:1 17:43, 30:40, 39:67.

[3]   For a detailed discussion on 'Tawhid' as the unifying harmonising principle in the Islamic worldview, see: Suddahazai (2015) PhD Thesis Chapter 4 'Islamic Worldview'.

[4]   As of 2015, there were five official Muslim Institutes of Higher Education in the UK. These included: Markfield Institute of Higher Education (Leicestershire); Cambridge Muslim College (Cambridgeshire); Muslim College (London); Institute for Ismailia Studies (London) (Suddahazai 2015).

[5]   The alim (male scholarship) and alima (female scholarship) qualifications are exclusively granted by traditional Muslim seminaries after students have successively undertaken and completed the traditional curriculums, which may take anywhere from 5–15 years. In the UK and subcontinent, this is normally the Deoband-derived Dars-e-Nizami curriculum. For more information: see Sikand (2005), Zaman (2002), Ebrahim (2015).

[6]   The two-year BA degree offered by MIHE in conjunction with Newman University is a unique program as it recognises the students' previous educational qualifications within the field of religious theology to be equivalent to the first year of a regular degree program and, therefore, the BA program at Markfield is only two years.

[7]   Courses: BISE 500/501; MF7 905; 906.

[8]   As a particular point of observation, I found male students to be more accepting of the traditional models and speak of their teachers in venerable terms as contrasted with the female students.

[9]   This was revealed by the students from the BA and MA courses during the 2nd session of the exercise described later in the paper.

[10]  The Oxford English Dictionary first recorded the term Pedagogy in 1571 AD.

[11]  "All the students" implies that every student at the MA and BA levels were able to identify with the prophet-teacher pedagogical and educational foundations of the Islamic worldview due to their Muslim heritage and knowledge of the Islamic faith ideology.

[12]  Recommended Reading: Ruther (2008).

13  For a more detailed discussion, see Lahmar (2020b).

14  A discussion of these factors is beyond the scope of this study. For a more detailed study, see Lahmar (2020b).

15  The conference referred to: I.H.K. Suddahazai (2021, December 14–15). 'Madrassa to Markfield: Transforming Modes of Religiosity Through Critical and Reflective Praxis' [Conference Session] H. Bashir & S. Mathee (Chairs), The Madrasah and the Modern Academy: Convergences, Departures and Possibilities for Collaboration [Symposium]; The Markfield Conference on Education, Leicestershire, United Kingdom.

16  All names used in this exercise are aliases and have been changed to ensure ethical integrity.

17  For more information, see: Asch (1956).

18  At the time of writing, Dr Haroon Bashir was Course Leader for the MA program in Islamic Studies and Director of Research at the Markfield Institute of Higher Education.

19  From Hadith Jāmi' Bayān al-'Ilm 1/53.

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
