# Peer review of "Reflecting on Teaching Practice: Adopting Islamic Liberatory Pedagogies within Muslim Institutes of Higher Education in UK (MIHEUK)"

_religions, doi:10.3390/rel14020223_

Round 1

Reviewer 1 Report

1. Line 90, footnote no. 4, refers to 2015 data, could it be more update data?

2. Talking about reflective pedagogy is not only talking about Freire's conscientization theory, but also improving cognitive abilities. A part of Muslim's tradition in pedagogy, as mentioned, is teacher centered in which students were asked to memorize and understand. These can be referred to lower cognitive skills, whereas reflection needs some higher levels of cognitive abilities, namely analysis, evaluation, and, at some points, innovation. the writer may want to include this in the analysis. 

Author Response

Point 1. Line 90, footnote no. 4, refers to 2015 data, could it be more update data?

Response 1: This data is from my doctoral thesis (2015) focusing upon Muslim hybrid Institutions for higher education in the UK. To date, there has been no further study on these institutions, the only reference made to these institutions is through the citation of my findings as recorded in my original thesis findings of 2015. The papers referencing my study were:

  • (Suddahazai 2021) SUDDAHAZAI, IHK (2021) “The Development of Leadership through Islamic Education: An empirical inquiry into religiosity and the styles of educational leadership experienced” International Journal of Education 3 (1) [DOI: https://doi.org/10.31763/ijele.v3i1.111]
  • (Arar et al 2022) ARAR, K., SAWALHI, R. and M. YILMAZ (2022) “The Research on Islamic-Based Educational Leadership since 1990: An International Review of Empirical Evidence and a Future Research Agenda” Religions 13 (42) DOI: 10.3390/rel13010042

Point 2. Talking about reflective pedagogy is not only talking about Freire's conscientization theory, but also improving cognitive abilities. A part of Muslim's tradition in pedagogy, as mentioned, is teacher centered in which students were asked to memorize and understand. These can be referred to lower cognitive skills, whereas reflection needs some higher levels of cognitive abilities, namely analysis, evaluation, and, at some points, innovation. the writer may want to include this in the analysis. 

Response 2: The reviewers’ advice to incorporate this much valued observation in the analysis section was duly conducted.

As the passage already addressed the lower cognitive abilities, the burden was to provide further light on the higher cognitive functions such as reflection, which formulates the essence of the paper. As the methodological concept of reflection had already been addressed, this passage has been reconstructed to briefly analyse the dearth in reflective development as a result of sole teacher centric approaches. Thereby, a short paragraph has been re-written into the passage from line [249].

This was also recorded using the MS tracking changes function.

The additional material consisted of the following content:

These are processes through which the learner develops ‘…the ability to reflect on the learning experience and incorporate new knowledge with pre-existing knowledge’ (Stoney & Oliver, 1999, p26). Dewey (1933) perceived this as an ‘active, persistent, and careful consideration of any belief or supposed form of knowledge in the light of the grounds that support it and the further conclusions to which it tends’ (p9). Thereby, reflection becomes an ‘inner dialogue, where the learner takes on both the protagonist and the antagonist roles sequentially in an attempt to reframe [their] understanding…[as] learners retreat from interaction to focus on individual and internal reflections’ (Soo & Bonk, 1998, p3-6). Thereby, making ‘interactions meaningful, leading to changes that are referred to as learning’. (Aminifar and Bahiraey, 2010, p414,) However, in contradistinction, traditional teacher-centred approaches condition the lower cognitive processes such as detection, response, remember and recall, to become more functional and automated (Mayer 2008; Tobias & Duffy 2009).

Reviewer 2 Report

Overall, the article shows strong merit with the author investigating an important pedagogical feature of Islam education in the third space between secular universities and the classical madrasa. I strongly recommend it for publication.

Author Response

There are no points to address from the reviewer. 

Reviewer 3 Report

I believe this is a well-written, argumentative essay. While nature is qualitative, the elucidation from the examples and its relevant literature makes it a compelling piece. 

Author Response

(The authors gave the same response as above.)
